# Driving Risk Assessment Using Near-Miss Events Based on Panel Poisson Regression and Panel Negative Binomial Regression

**DOI:** 10.3390/e23070829

**Published:** 2021-06-29

**Authors:** Shuai Sun, Jun Bi, Montserrat Guillen, Ana M. Pérez-Marín

**Affiliations:** 1Key Laboratory of Transport Industry of Big Data Application Technologies for Comprehensive Transport, School of Traffic and Transportation, Beijing Jiaotong University, Beijing 100044, China; sunshuai@bjtu.edu.cn; 2Department of Econometrics, Riskcenter-IREA, Universitat de Barcelona, 08034 Barcelona, Spain; amperez@ub.edu

**Keywords:** driving risk assessment, usage-based insurance, driving risk score, telematics, near-miss event, driving behavior, panel data analysis, count data model, econometrics, generalized linear model

## Abstract

This study proposes a method for identifying and evaluating driving risk as a first step towards calculating premiums in the newly emerging context of usage-based insurance. Telematics data gathered by the Internet of Vehicles (IoV) contain a large number of near-miss events which can be regarded as an alternative for modeling claims or accidents for estimating a driving risk score for a particular vehicle and its driver. Poisson regression and negative binomial regression are applied to a summary data set of 182 vehicles with one record per vehicle and to a panel data set of daily vehicle data containing four near-miss events, i.e., counts of excess speed, high speed brake, harsh acceleration or deceleration and additional driving behavior parameters that do not result in accidents. Negative binomial regression (
$AIC_{overspeed}$
 = 997.0, 
$BIC_{overspeed}$
 = 1022.7) is seen to perform better than Poisson regression (
$AIC_{overspeed}$
 = 7051.8, 
$BIC_{overspeed}$
 = 7074.3). Vehicles are separately classified to five driving risk levels with a driving risk score computed from individual effects of the corresponding panel model. This study provides a research basis for actuarial insurance premium calculations, even if no accident information is available, and enables a precise supervision of dangerous driving behaviors based on driving risk scores.

## 1. Introduction

Near-miss events are incidents that denote the existence of danger, even if no accident occurs. Reporting of near-miss events is an established error reduction technique that has been used by many industries to manage risk and reduce accidents. In the auto insurance industry, insurers traditionally calculate premiums by analyzing past claims reported by the insured policy holders, and reward those drivers that do not report accidents with a no-claims bonus. However, this may be a rather incorrect approach to the assessment of accident risk, especially when the insured has suffered accidents but chooses not to make a claim so as not to lose the no-claims bonus. Fortunately, the advent of the Internet of Vehicles (IoV) offers a better solution to this problem, using near-miss events to identify driving risk. Near-miss events ultimately provide information that can lead to actuarial premium calculations in the auto insurance industry [1,2].

This study explores how to evaluate driving risks, in the short term, and to score drivers without claims and accidents based on information on near-miss counts over a short period of time. One of the main novelties of this approach, in the absence of claims, is to use telematics sensors for observation of drivers over a given period. The model obtained in this study offers an important alternative for driving risk identification. Not only can the model reflect risk factors that influence each near-miss event but it can also help to evaluate drivers’ risks, and fixed-effects panel count data models can be used to rank drivers according to their individual effects. The modeling method and results are invaluable for insurance companies for developing usage-based insurance (UBI) to personalize premiums. They are also of interest to traffic regulatory authorities for promoting safe driving and the prevention of accidents.

Near-miss events are incidents that need to be defined and extracted from the original raw data files for further processing and analysis. By dealing only with near-miss events, and excluding claims or accidents, this study aims to specifically identify driving patterns. This study is carried out both on a per driver summary data set and on a panel data set where a daily summary is shown for each driver. Our data contain counts of the four types of near-miss events in our study. Speeding, high speed braking, harsh acceleration and harsh deceleration have been defined based on actual driving conditions and local laws and regulations. Other high-risk events, e.g., sharp turning, dangerous lane changing and unexpected maneuvers, proved by previous studies to be related to driving risk, are not included in this study due to the dimension and precision limitations of the original data set.

Our interest is to model the frequency of near-miss events given the drivers’ characteristics. The simplest statistical model that links a count data dependent variable with explanatory factors is the Poisson model. Essentially, the Poisson model is similar to linear regression, where a response depends on some others inputs. Here we think that distance driven or mean speed among others, influence the expected frequency of near-miss events. A Poisson model, which is also known as a Poisson regression model, is easily interpretable and provides a way to elucidate the significant effects on the conditional expected frequency. Poisson models are constrained by the fact that conditional expectation and conditional variance are equal. Negative binomial regression models are a natural extension that overcomes this restriction. More details on the models are provided in the Methods section below.

Since the extracted frequency of near-miss events is an unbounded non-negative integer, Poisson regression and negative binomial regression are both suitable for modelization. Poisson regression, negative binomial regression, zero-inflated Poisson regression and zero-inflated negative binomial regression are respectively applied to the summary data set. Average speed, brake times, accelerator pedal position, engine fuel rate etc., are selected as independent variables. Either mileage or fuel consumption can be chosen as the exposure variable to offset the model. In order to reach a clear understanding of risk factors of different near-miss events, each near-miss event is individually used as a dependent variable. However, regardless of which one is selected as the dependent variable, negative binomial regression is shown to provide the best fit in the summary data in this study.

Negative binomial regression also performs better than Poisson regression on the panel data sets. Individual effects and time effects are estimated using panel Poisson regression and panel negative binomial regression on a short panel data set of six days in length. The regression results confirm the existence of individual effects and time effects, and also enable the driving risk of each vehicle to be ranked. The driving risk level of vehicles can then be classified by converting the individual effects into scores, thus providing an important reference for further accurate calculation of premiums.

The rest of this article is organized as follows. The development of UBI and previous efforts on driving risk assessment are summarized in Section 2. Section 3 describes the data and introduces the key parameters used in modeling. Section 4 presents the model expression of Poisson regression and negative binomial regression used in the study. The results of negative binomial regression using the summary data set and the panel data set are reported and analyzed in Section 5. The results are discussed and the conclusions are presented in Section 6.

## 2. Literature Review

The auto insurance industry is continuously pursuing new ways to calculate more accurate actuarial premiums. However, traditional auto insurance calculations are limited by the difficulty of obtaining information on policy holders, so classical ratemaking uses simple information on drivers (age gender,), vehicles (type of car, model and brand) and driving sections [3]. With current advances in information technology, a new type of insurance business, UBI, based on multi-source data and personalized premium calculation is becoming the mainstream. The Pay-as-you-drive (PAYD) mode of charging premiums is based on mileage or fuel consumption, on the premise that mileage or fuel consumption correlates with the probability of suffering an accident [4]. PAYD has evolved into a newer scheme, called the pay-how-you-drive (PHYD) ratemaking mode, which is based on multiple sources of data, including driving behavior data [5]. Following the development of 5G communication technology, it may now be possible to implement an even more sophisticated monitoring and pricing strategy, known as the manage-how-you-drive (MHYD) principle, i.e., real-time calculation of premiums based on multi-source data and providing real-time information to drivers to restrain from bad driving behavior [3,6]. However, due to technological, regulatory and other issues regarding privacy [7], there is still no mature PHYD product on the market at present [8,9] and, in terms of MHYD, further research is necessary on driving risk to produce products that better reflect the driver profile [10].

Traffic accidents all over the world result in a large number of casualties every year, and high-risk driving is one of the main factors behind these incidents [3]. Consequently, research on driving risk has been a topic of interest over recent decades. Simulation experiments to evaluate driving risk have been designed in the laboratory setting to identify driving risk factors [11,12,13,14] as well as experiments using actual vehicles on the road [15,16,17,18,19]. Questionnaire surveys for driving risk assessment have also been studied [20,21]. In fact, the naturalistic type of driving data collected by the IoV or smart phones, known as telematics data, can effectively reduce the influence of subjective factors and unreasonable assumptions in producing effective risk-mitigating actions [22,23,24,25,26].

In research related to driving risk assessment in the auto insurance industry, machine learning and generalized linear models feature equally. Machine learning, with its strong ability to process big data efficiently, is increasingly gaining ground in its application in the auto insurance business. Logistic regression [27], cluster analysis [28], decision tree [5], support vector machine [29], neural network [30] and other machine learning models [31,32,33] have been widely studied in the field of driving risk assessment, and the results have shown machine learning to be a powerful tool [34]. However, since most machine learning procedures, being black box algorithms, do not offer a high degree of interpretability, they cannot completely replace the conventional generalized linear models implemented for decades in the auto insurance industry [8].

Conventional generalized linear models discern the correlation between influencing factors and claims or accidents in frequency and severity models [9,24,25,35]. However, the study of near-miss events even when there is a lack of information on claims and accidents should not be ignored [2,15]; on the contrary, since near-misses are more frequent than accidents and are positively associated with them, they can be considered a good alternative for risk modeling for driving risk assessment [1]. Compared with previous studies, this study not only conducts regression on the summary data set to model and analyze the factors causing near-miss events, but also conducts panel data regression on the panel data set to consider individual effects and time effects. The regression results can not only make more accurate causal inference, but also carry out risk scoring.

## 3. Data Description

The telematics data used in this study are collected from an IoV information service provider in China. While we cannot obtain more data due to the commercial privacy of the data, the limited data also contains valuable driving risk information, which is worth studying. The original data set contains 182 data files, representing sensor data for 182 vehicles observed from 3–8 July 2018 [10]. Each data file contains 62 different measurements but, after data processing [36], less than one-third of them can be used due to recording errors and inconsistencies. The original data are transformed for modeling into a summary data set with information on each driver (see details in Table 1).

The variables overspeed, highspeedbrake, harshacceleration and harshdeceleration are individually filtered by combining the rules of traffic law and driving code. Previous studies have confirmed that speeding is a dangerous driving behavior which is likely to cause traffic accidents [3]. In China, traffic safety regulations stipulate a maximum speed for each type of vehicle on all types of roads. The maximum speed limit for the vehicles in this study is 90 km/h; exceeding this by 
10%
 is not deemed to be a traffic offense. Therefore, 100 km/h is taken as the threshold value of the overspeed near-miss event. Another high risk near-miss event that deserves attention is that of emergency braking; at high speed (>90 km/h), if the brake is not used correctly or is subjected to lateral force, the car is prone to side-slip or even cartwheel. Lastly, both harsh acceleration and harsh deceleration are near-miss events that compromise driving safety and fuel economy. Based on previous research experience [1,2,37] and the filter analysis of the extreme values of this data set by box graph method, 6 m/s
${}^{2}$
 is determined as the filtering threshold value of harsh acceleration and harsh deceleration. Figure 1 shows that near-miss events are all non-negative integers. Combined with the relationship between expectation and variance shown in Table 1, the four near-miss events are shown to be suitable as dependent variables of a Poisson regression or a negative binomial regression.

The panel data set has one summary per day for each driver. The statistics of the panel data set are shown in Table 2.

## 4. Methods

Poisson regression is a generalized linear model. Negative binomial regression can be considered as a generalization of Poisson regression with overdispersion of the dependent variable 
$Y_{i}$
 where subindex i refers to the i-th observation in the data set. The probability density function of the Poisson distribution is:
(1)
$$P\left(Y_{i}=y_{i}|x_{i}\right)=\frac{e^{-\lambda_{i}}\lambda_{i}^{y_{i}}}{y_{i}!}$$

where 
$\lambda_{i}$
 is the Poisson arrival rate and is determined by explanatory variable 
$x_{i}$
 in Poisson regression to represent the average number of events, which is equal to the expectation and variance of the explained variable 
$E\left(Y_{i}|x_{i}\right)=V\left(Y_{i}|x_{i}\right)=\lambda_{i}$
.

The negative binomial distribution is a mixture of a Poisson (
λ
) and a Gamma (*a*,*b*) distribution. The probability density function of the negative binomial distribution is:
(2)
$$f\left(y|a,b\right)=\int_{0}^{\infty }f\left(y|\lambda \right)g\left(\lambda |a,b\right)d\lambda =\frac{\Gamma (y+a)}{\Gamma (y+1)\Gamma (a)}{\left(\frac{b}{1+b}\right)}^{a}{\left(\frac{1}{1+b}\right)}^{y}$$

where 
λ
 is the mean and variance of the Poisson distribution, *a* is the shape parameter of the Gamma distribution, *b* is the inverse scale parameter of the Gamma distribution, 
$E\left(y\right)=\frac{a}{b}=\bar{\lambda }$
 and 
$V\left(y\right)=\frac{a}{b}\left(1+\frac{1}{b}\right)=\bar{\lambda }\left(1+\frac{\bar{\lambda }}{a}\right)$
.

The zero-inflated model is applicable when the counting data contains a large number of zero values. Theoretically, it is a two-stage decision. First, it decides whether to choose zero or a positive integer, and then it determines which positive integer to choose. Therefore, the probability distribution of 
$Y_{i}$
 is a mixed distribution:
(3)
$$Pr\left(Y_{i}=y_{i}|x_{i}\right)=\left\{\begin{array}{cc} \theta +\left(1-\theta \right)P\left(K_{i}=y_{i}|x_{i}\right) & y_{i}=0\\ \left(1-\theta \right)P\left(K_{i}=y_{i}|x_{i}\right) & y_{i}>0 \end{array}\right.$$

where 
θ
 is the probability of an extra zero value, 
$K_{i}$
 can follow a Poisson distribution or a negative binomial distribution depending on the characteristics of the dependent variable.

The conditional expectation function of a negative binomial regression model depends on a vector of explanatory variables 
$x_{i}$
 and, similar to Poisson, is usually defined by a log-link as:
(4)
$$E\left(y_{i}|x_{i}\right)=\lambda_{i}=T_{i}\times \exp\left(\alpha +\beta_{1}x_{1i}+\cdots +\beta_{k}x_{ki}\right)$$

where *i* is the number of the observation, *k* depends on the number of independent variables, 
$T_{i}$
 denotes the offset variables (so, in our application, 
$kilo_{i}$
 or 
$fuel_{i}$
 is the exposure variable), 
$x_{1i}$
...
$x_{ki}$
 represent the independent variables such as 
$brakes_{i}$
, 
$range_{i}$
, 
$speed_{i}$
, 
$rpm_{i}$
, 
$acceleratorpedalposition_{i}$
 and 
$enginefuelrate_{i}$
, 
α
 and 
β1
...
$\beta_{k}$
 are unknown parameters that need to be estimated.

The two-way fixed effect model of panel Poisson regression and panel negative binomial regression is specified as:
(5)
$$E\left(y_{it}|x_{it}\right)=\lambda_{it}=T_{it}\times \exp\left(\alpha +\beta_{1}x_{1it}+\cdots +\beta_{k}x_{kit}+d_{i}+p_{t}\right)$$

where *i* is the number of the observation, *t* is of time reference, *k* depends on the number of independent variables, 
$T_{it}$
 is the offset and equals 
$kilo_{it}$
 or 
$fuel_{it}$
 as the exposure variable of the *i*th observation at time *t*, 
$x_{1it}$
...
$x_{kit}$
 represent the independent variables of the *i*th observation at time *t* such as 
$brakes_{it}$
, 
$range_{it}$
, 
$speed_{it}$
, 
$rpm_{it}$
, 
$acceleratorpedalposition_{it}$
 and 
$enginefuelrate_{it}$
, 
α
 and 
β1
...
$\beta_{k}$
 are unknown parameters that need to be estimated, 
$d_{i}$
 represents the individual effect and 
$p_{t}$
 represents the time effect. To avoid identification problems in the model specification, 
$d_{1}=p_{1}=0$
.

The methodology of this study involves data preparation, modeling, risk scoring of driving risk, etc. The whole technical process is shown in Figure 2.

In the data preparation stage, the original data need to be preprocessed, including multi-source data fusion, data cleaning, missing processing, etc. Then the summary data set and panel data set required in this study are obtained through statistical calculation. In the modeling phase, multiple count data models are used on two data sets for regression analysis, which follows certain premises. Our observed drivers can be considered independent of each other. Even if they drive in a similar area, they do not have any apparent relationship between each other. When we observe one driver over time, we have taken care of temporal correlation using the panel model that considers that one individual is observed repeatedly, here each day. In the scoring stage, the regression results obtained from the regression model most suitable for the data in this study can be used for causal analysis of near-miss events and driving risk scoring and rating. In the research field of telematics data application, the results of this study show this application has potential in, for example, driving behavior supervision and personalized premium calculation. The work at this stage has yet to be completed. Data processing in the preparation and Poisson regression and negative binomial regression on different data sets in the modeling can be implemented with data tools such as Stata, Python, R, etc.

## 5. Results

Before regression, multicollinearity tests are carried out on all explanatory variables to eliminate the influence of multicollinearity on the model. As shown in Table 3, the variance inflation factors (VIF) of all selected independent variables are less than 5, while the correlation coefficients are generally less than 0.7. This indicates that the multicollinearity among variables is weak, so all of them can be included in the regression equation and robust estimates can be made.

Both Poisson regression and negative binomial regression are applicable to this study, and the zero-inflated model is taken as a consideration for the large number of zero values of dependent variables. In order to determine the regression model which is most suitable for this study, the performance of the two models on different dependent variables is compared. All the estimated results are obtained by regression after standardization of the original values.

### 5.1. Results of the Summary Data Set

In the summary data set, four near-miss events are respectively treated as dependent variables while the independent variables are brakes, speed, rpm, accelerator pedal position and engine fuel rate, where kilo is chosen as the exposure variable or offset. Poisson regression, zero-inflated Poisson regression, negative binomial regression and zero-inflated negative binomial regression are estimated (see Table 4). Regardless of which near-miss event is the dependent variable, negative binomial regression has maximum log-likelihood value, and minimum AIC value and BIC value. That is, negative binomial regression has the best performance in this data set.

According to the results of negative binomial regression in different dependent variables (see Table 5 and Figure 3a), different near-miss events are affected by different driving risk factors with different influences. Overall, the average speed has the most obvious influence on near-miss events, with a significant negative effect on harsh acceleration (−0.776) and harsh deceleration (−0.658). The impact of braking event number on near-miss events is also positive significant. The higher the number of braking, the more high speed braking (0.272), harsh acceleration (0.189) and harsh deceleration (0.180) occur. In addition, average RPM is positively correlated with harsh acceleration (0.178), and average accelerator pedal position is positively correlated with harsh acceleration (0.152) and harsh deceleration (0.235). Interestingly, some influencing factors have opposite effects on different dependent variables. Range of driving has a positive effect on high speed brake (0.272) but a negative effect on harsh deceleration (−0.153) while average engine fuel rate has a significant positive effect on high speed braking (0.705) but a negative effect on sharp deceleration (−0.157). Furthermore, the significance of the constant term indicates that, in addition to the factors considered in this study, there are other factors that also influence near-miss events. The results of the other three regression models on the summary data set are shown in Table A1, Table A2 and Table A3, and discussed in the Discussion section.

### 5.2. Results of the Panel Data Set

As shown in Table 6, the evaluation index (log-likelihood, AIC and BIC) of negative binomial regression is lower than that of Poisson regression for each dependent variable. Therefore, negative binomial regression is better than Poisson regression on panel data.

The panel negative binomial regression is used to estimate the two-way fixed effect model, considering both individual effect and time effect on four dependent variables. The influencing factors reflected by this (see Table A4 and Figure 3b) differ from those shown in the results of the summary data. For example, harsh acceleration and harsh deceleration are positively affected by the number of brakes (0.246 and 0.253) and average accelerator pedal position (0.249 and 0.270) but negatively affected by the average speed (−0.645 and −0.586) and average engine fuel rate (−0.188 and −0.229). However, RPM, which is not significant in the summary data, is significantly positive for overspeed (1.683) and high speed braking (1.287). The brakes (0.0505) and engine fuel rate (0.295), which had a significant positive effect on the summary data, become insignificant.

The advantage of panel data over summary data is that fixed effects can be estimated and thus individual effects and time effects can be interpreted. The time effect is significant in most cases for high speed braking, harsh acceleration and harsh deceleration, which indicates that these three near-miss events are greatly influenced by time. The time effect on the overspeed event is significant for only one day, suggesting that it is less influenced by time. More importantly, the individual effects of the four near-miss events can be used to score each observation. It should be noted that the first observation has been omitted in the regression to avoid complete multicollinearity, and its value is expected to be zero in the subsequent driving risk score.

## 6. Discussion

The regression results of Poisson regression (see Table A1), zero-inflated Poisson regression (see Table A2), negative binomial regression (see Table 5) and zero-inflated negative binomial regression (see Table A3) on the summary data set show the importance of driving behavior variables in driving risk. The high significance of two variables, braking times and average speed, in the four regression models indicates that these two factors have a very important impact on the generation of near-miss events. Moreover, the significant performance of specific independent variables in the regression model of specific dependent variables indicates that near-miss events are affected by a variety of driving behavior factors and the formation mechanism of each near-miss event is different. For example, the positive effect of RPM on harsh acceleration events, the positive effect of accelerator pedal position on harsh deceleration events and the positive effect of engine fuel rate on high speed braking events are shown in Table A1, Table A2 and Table A3, Table 5.

The results obtained by panel regression are more reliable than those obtained by pooled regression. Table 5 and Table A4 and Figure 3 show that some coefficients that are not significant in the pooled negative binomial regression become significant in the panel negative binomial regression, while some significant parameters in the pooled negative binomial regression are not significant in the panel negative binomial regression. This means that the dependent variables are affected by individual effects and time effects. In the panel negative binomial regression, most of the individual and time coefficients are significant, which indicates the suitability of this type of regression analysis.

Driving risks can be evaluated by the regression coefficients of negative binomial models on panel data. The value of the individual coefficients within a regression indicates the individual’s deviance from the level of the expected occurrence of a particular near-miss event, given the information on all the other explanatory variables. In other words, the individual effect coefficient can be understood as the effect utility of each vehicle on the occurrence of the corresponding near-miss event. Geometrically, the effect coefficient of each individual is a change in the intercept.

Four near-miss events are used as dependent variables to obtain four sets of regression coefficients. Given that the influencing factors and generating mechanisms of different near-miss events are different, combining the four groups of regression coefficients into one group is not recommended. However, harsh acceleration and harsh deceleration show very similar characteristics in terms of data description before regression (Table 1 and Table 5), after regression (Figure 3 and Table A4) and in distribution of driving risk score (Table A5). Even so, it is not recommended to combine them into a single near-miss event for study, because the occurrence conditions and coping operations of them are different, and it is the most appropriate choice to study each near-miss event separately.

In order to transform individual effect estimates of near-miss models into a driving risk grading, several steps need to be followed. Firstly, winsorization avoids the influence of possibly spurious outliers (the double tail was winsorized with the threshold 0.01 in this study). Secondly, the regression coefficient can be compressed to the interval of [0,1] through normalization. Each group of coefficients is then mapped into an interval of [0,5] (see Table A5), and each observation then is given a driving risk level from 1 to 5, i.e., excellent, good, medium, bad and terrible (see Figure 4). The values of exactly 0 and 5 are included because the corresponding observations are the minimum and the maximum values in their group and are Min-Max scaled. In *overspeed* and 
highspeedbrake
 groups, two types of observations with high risk or low risk can be clearly seen. This indicates that these two near-miss events are more sensitive to driving behavior than 
harshacceleration
 and 
harshdeceleration
 and can be considered as a higher priority and weight in subsequent studies. Note that the same observation (id125) has different risk levels for different near-miss events, which also explains why multiple near-miss events cannot be analyzed together. Ultimately, the premium would be charged individually according to the driving risk level of the insured person.

## 7. Conclusions

The number and type of dependent variables and independent variables selected in this study are limited by the size and quality of the original data. With the promotion and innovation of IoV and of new energy vehicles, the amount and dimension of data will be greatly increased. Therefore, application of near-miss events as dependent variables could be easily increased or decreased, according to needs. For example, sharp turn should be included, if possible, as a near-miss event because sharp turn is a highly studied and accident-proven pattern of high driving risk. For the same reason, more driving behavior indicators, such as steering wheel angle speed, brake pedal position, and so on, could be used as independent variables in the regression model. In addition, traditional auto insurance factors, such as driver information, vehicle information, road information, environment information and the health status of batteries (of new energy vehicles) should be considered to provide more optional independent variables for the model.

In practical applications, near-miss events can be combined with claims and accidents to accurately evaluate driving risks. This study proves that near-miss events can be used as driving risk scores when there are no claims or accidents. However, when claims or accidents exist, the driving risk score obtained from claims or accidents can be used as the basis for premium calculation, while the driving risk rating obtained from near-miss events can be used to remind and warn drivers to reduce the corresponding dangerous driving habits.

In this study, the best performing negative binomial regression (see Table 4) was selected as the main method for modeling on our data set. The model is suitable for similar causal analysis of similar data sets. However, in case of risk event prediction or analysis on other data sets, it is necessary to reevaluate the goodness of fit of various models, and even machine learning methods with good prediction performance should be taken into consideration. The optimal method is not fixed, but depends on the data, conditions and purposes.

Econometrics and machine learning complement each other. The generalized linear model established in this study reveals the relationship between driving behavior factors and near-miss events, and gives a driving risk score for each observation. This model has strong explanatory power, but its generalization degree and robustness need to be further tested, especially on larger data volume and data dimension. The successful application of machine learning methods in many fields shows that they are often effective in dealing with big data problems but that their results cannot always be easily interpreted, and this interpretation is exactly what the insurance field values. Therefore, telematics data application offers a new way to help find a balance between econometrics and machine learning so as to have good explainability, good generalization ability, quick response ability, and so on [38,39].

In general, near-miss events can provide insurers with effective risk information in the absence of claims and accident data. In our real case study, negative binomial regression is the most suitable modeling method for near-miss events as dependent variables. This study provides a technical reference for the promotion and development of PHYD ratemaking schemes. 

## Figures and Tables

**Figure 1 entropy-23-00829-f001:**
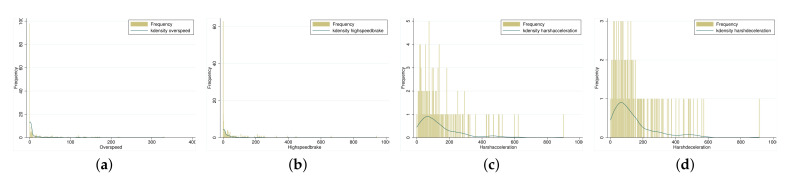
Histogram of frequency distribution of four near-miss events: (**a**) Over speed; (**b**) High speed brake; (**c**) Harsh acceleration; (**d**) Harsh deceleration.

**Figure 2 entropy-23-00829-f002:**
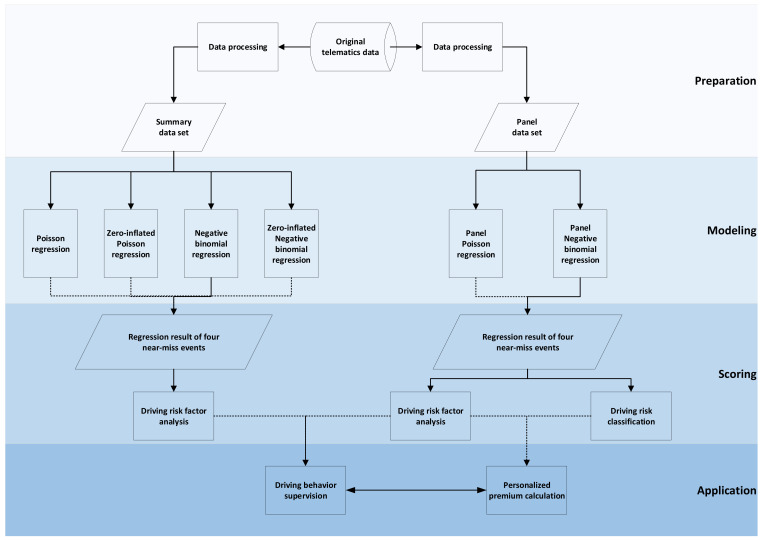
Technical flow chart.

**Figure 3 entropy-23-00829-f003:**
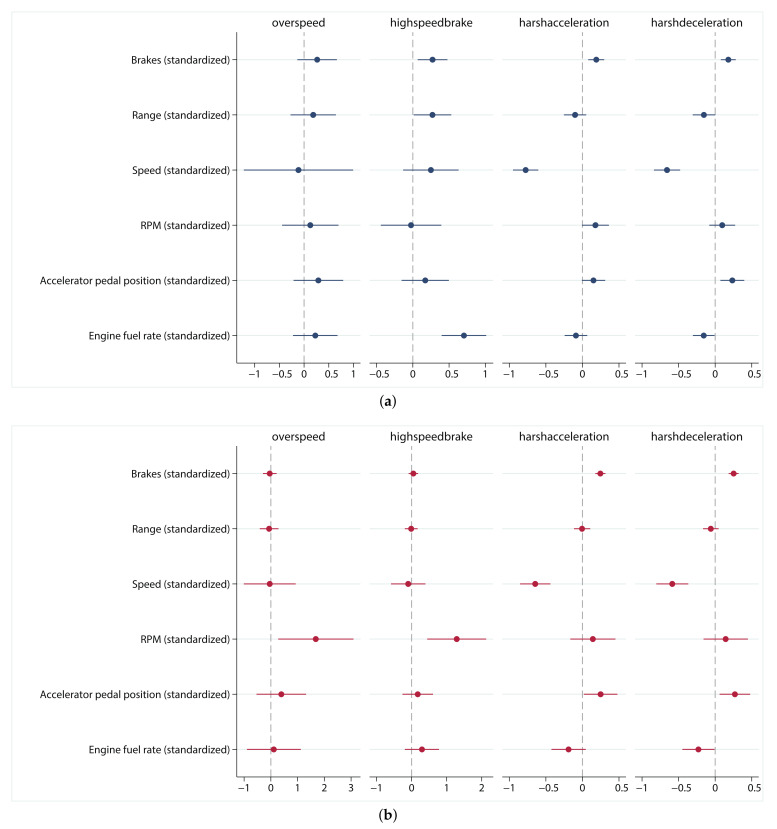
Partial coefficient estimation results of (**a**) negative binomial regression; (**b**) Panel negative binomial regression.

**Figure 4 entropy-23-00829-f004:**
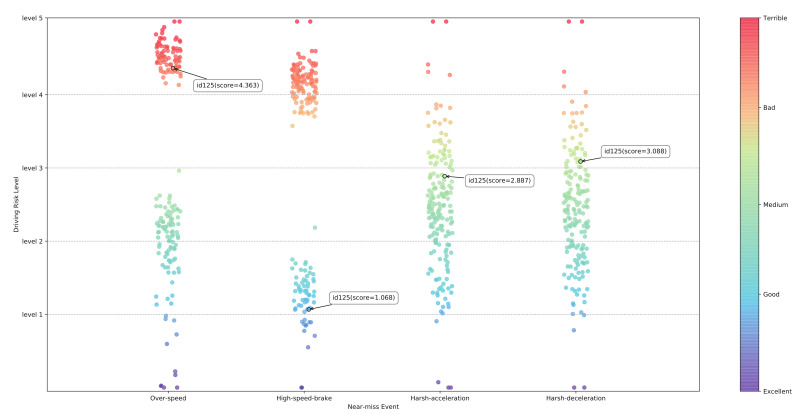
Driving risk ranking of four near-miss events.

**Table 1 entropy-23-00829-t001:** Descriptive statistics of the summary data set for 182 drivers observed from 3–8 July 2018.

Variable	Mean	Standard Deviation	Minimum	Median	Maximum	Defination
overspeed	19.19	45.37	0	0	330	Frequency of driving speed greater than 100 km/h
highspeedbrake	44.23	108.3	0	4	942	Frequency of braking when the driving speed is greater than 90 km/h
harshacceleration	139.0	134.7	0	101	899	Frequency of cases when acceleration is greater than 6 m/s ${}^{2}$
harshdeceleration	141.9	137.8	1	105	913	Frequency of cases when acceleration is less than 6 m/s ${}^{2}$
kilo	2223	1674	3.73	1832.175	7164	Total driving distance (km)
fuel	621.7	470.9	10.25	487.295	2018	Total fuel consumption (L)
brakes	1588	1426	6	1138.5	9243	Total number of brakes
range	5.201	5.021	0.027	3.399	26.78	Range of driving (geographical units)
speed	36.88	16.37	0.297	36.657	67.84	Mean of speed (km/h)
rpm	1028	188.3	233.1	1009.301	1620	Mean of revolutions per minute (r/min)
acceleratorpedalposition	21.05	7.110	0.187	21.26	39.29	Mean of acceleration pedal position (%)
enginefuelrate	11.52	4.464	1.868	11.203	22.01	Mean of engine fuel rate (%)

The number of each parameter is 182.

**Table 2 entropy-23-00829-t002:** Descriptive statistics of a panel data set for 182 drivers observed over six days (total cases 1092).

Variable	N	Mean	Standard Deviation	Minimum	Median	Maximum
overspeed	1092	3.199	14.37	0	0	315
highspeedbrake	1092	7.435	21.74	0	0	215
harshacceleration	1092	23.37	29.78	0	14	223
harshdeceleration	1092	23.86	30.16	0	13.5	233
kilo	1092	372.6	373.2	0	263.24	1739
fuel	1092	104.1	105.7	0	72.15	565.8
brakes	1092	264.7	291.0	0	178	1940
range	1092	2.406	2.963	0	1.243	14.07
speed	1092	31.96	21.58	0	31.514	77.74
rpm	1092	894.3	346.9	0	973.714	1731
acceleratorpedalposition	1092	17.51	10.19	0	18.613	45.74
enginefuelrate	1092	9.794	5.835	0	10.018	26.18

**Table 3 entropy-23-00829-t003:** Variance inflation factor and correlation of explanatory variables.

Variable	VIF	Brakes	Range	Speed	rpm	Accelerator Pedal Positon
brakes	3.07					
range	2.65	0.1213				
speed	2.30	0.0536	0.6262			
rpm	2.13	−0.0254	−0.0203	0.1804		
accelerator pedal positon	2.03	0.0154	0.1174	0.3458	0.7695	
engine fuel rate	1.04	0.1687	0.6313	0.6490	0.1075	0.3529

**Table 4 entropy-23-00829-t004:** Model performances of Poisson, zero-inflated Poisson, negative binomial and zero-inflated negative binomial in summary data set.

Variable	Model	N	Log-Likelihood	df	AIC	BIC
overspeed	POS	182	−3518.92	7	7051.846	7074.274
	ZIP	182	−2369.82	8	4755.64	4781.272
	NB	182	−490.517	8	997.0338	1022.666
	ZINB	182	−490.516	9	999.0315	1027.868
highspeedbrake	POS	182	−2830.75	7	5675.498	5697.926
	ZIP	182	−2667.02	8	5350.034	5375.666
	NB	182	−627.422	8	1270.843	1296.476
	ZINB	182	−627.422	9	1272.843	1301.68
harshacceleration	POS	182	−5857.26	7	11,728.51	11,750.94
	ZIP	182	−5857.26	8	11,730.51	11,756.14
	NB	182	−1032.81	8	2081.623	2107.255
	ZINB	182	−1032.81	9	2083.623	2112.459
harshdeceleration	POS	182	−6269.47	7	12,552.93	12,575.36
	ZIP	182	−6269.47	8	12,554.93	12,580.56
	NB	182	−1037.14	8	2090.285	2115.917
	ZINB	182	−1037.14	9	2092.285	2121.121

**Table 5 entropy-23-00829-t005:** The results of negative binomial regression for four near-miss events in the summary data set of drivers.

Variable	Overspeed		Highspeedbrake		Harshacceleration		Harshdeceleration	
	Coefficient	z	Coefficient	z	Coefficient	z	Coefficient	z
constant	−5.175 ***	−15.87	−5.114 ***	−35.36	−2.548 ***	−43.39	−2.525 ***	−42.99
brakes	0.264	1.29	0.272 **	2.60	0.189 ***	3.38	0.180 ***	3.45
range	0.185	0.79	0.272 *	2.05	−0.100	−1.29	−0.153	−1.94
speed	−0.113	−0.20	0.249	1.28	−0.776 ***	−8.81	−0.658 ***	−7.20
rpm	0.125	0.43	−0.0241	−0.11	0.178	1.90	0.0969	1.07
acceleratorpedalposition	0.290	1.13	0.171	1.03	0.152	1.87	0.235 **	2.82
enginefuelrate	0.227	0.99	0.705 ***	4.49	−0.0883	−1.12	−0.157 *	−2.07
log-likelihood	−490.5		−627.4		−1032.8		−1037.1	
AIC	997.0		1270.8		2081.6		2090.3	
BIC	1022.7		1296.5		2107.3		2115.9	
Observation	182		182		182		182	

*** *p* < 0.01, ** *p* < 0.05, * *p* < 0.1.

**Table 6 entropy-23-00829-t006:** Model performances of Poisson and negative binomial in the panel data set of drivers with six observations per driver.

Variable	Model	N	Log-Likelihood	df	AIC	BIC
overspeed	XTPOS	1092	−1926.78	188	4229.559	5168.763
	XTNB	1092	−957.497	189	2292.993	3237.193
highspeedbrake	XTPOS	1092	−2594.37	188	5564.733	6503.937
	XTNB	1092	−1527.05	189	3432.105	4376.305
harshacceleration	XTPOS	1092	−6117.44	188	12,610.89	13,550.09
	XTNB	1092	−3526.09	189	7430.186	8374.386
harshdeceleration	XTPOS	1092	−6042.02	188	12,460.03	13,399.24
	XTNB	1092	−3547.66	189	7473.311	8417.51

## Data Availability

This study is based on the original telematics data from China Satellite Navigation and Communications Co., Ltd., which could not be made public due to confidentiality agreements.

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
