# Peer review of "Driving Risk Assessment Using Near-Miss Events Based on Panel Poisson Regression and Panel Negative Binomial Regression"

_entropy, 2021, doi:10.3390/e23070829_

Round 1

Reviewer 1 Report

This article, although interesting has some serious flaws in terms of statistical analysis and interpretation of the results. The overall presentation also needs to be improved a little. For example, there is a separate section for the literature review, which should be embedded into the introduction section. There are several other major concerns which are listed as follows:

[1] The authors need to justify why they have considered Poisson and Negative binomial regression, especially for folks from non-statistical backgrounds. Some generic descriptions are given but that is not sufficient.

[2] Introduction, page 2, line 34, .."the coefficients...." what are the coefficients?

[3] Page 6, Lines 170-174, Eq. (5)---several key aspects related to Poisson and Panel negative binomial regressions have been ignored, a non-exhaustive list of such items are listed below:

(a) Specific interpretation/explanation on each of the slope/regression coefficients \beta_{j}'s need to be provided.

(b) Are there any evidence of multicollinearity among the  \beta_{j}'s, i.e., with the predictors? If it is how it is tackled? Furthermore, there might be one or more lurking variables for which a dependency structure can be observed. Some discussion in this direction is necessary.

(c) Finally, are all the assumptions and/or necessary and sufficient conditions to conduct a Poisson and/or Negative binomial regression are met? 

(d) Page 7, Table 3: For each regression (best-fitted model), except for Negative binomial regression, nothing is mentioned on the efficacy of the slope coefficients!  The overall goodness of fit is not sufficient enough, we need to examine the role of each predictor. Some discussion in this direction is necessary.

Based on the above, I recommend a major revision of this paper.

Author Response

Dear Mr. or Ms.,

Thank you for your constructive suggestion for our flawed manuscript. We have undertaken major revisions and taken all your precious comments into consideration. Please find our detailed replies to the specific comments below.

Q1: The authors need to justify why they have considered Poisson and Negative binomial regression, especially for folks from non-statistical backgrounds. Some generic descriptions are given but that is not sufficient.

A1: We agree that we did not justify why we have considered Poisson and Negative binomial regression. The following sentences from the introduction have been modified from Line 50 to Line 60:

“Our interest is to model the frequency of near-miss events given the drivers’ characteristics. The simplest statistical model that links a count data-dependent variable with explanatory factors is the Poisson model. Essentially, the Poisson model is similar to linear regression, where a response depends on some others inputs. Here we think that distance driven or mean speed among others, influences the expected frequency of near-miss events. A Poisson model, which is also known as a Poisson regression model, is easily interpretable and provides a way to elucidate the significant effects on the conditionally expected frequency. Poisson models are constrained by the fact that conditional expectation and conditional variance are equal. Negative binomial regression models are a natural extension that overcomes this restriction. More details on the models are provided in the Methods section below.”

Q2: Introduction, page 2, Line 34, "the coefficients...." what are the coefficients?

A2: We have removed the word “the coefficients” because these are going to be specified when the model is defined. The sentence is clear because we refer to the model proposed in this work.

Page 6, Lines 170-174, Eq. (5) --several key aspects related to Poisson and Panel negative binomial regressions have been ignored, a non-exhaustive list of such items are listed below:

Q3: Specific interpretation/explanation on each of the slope/regression coefficients \beta_{j}'s need to be provided.

A3: We added the corresponding explanation from Line 189 to Line 191.

Q4: Are there any evidence of multicollinearity among the \beta_{j}'s, i.e., with the predictors? If it is how it is tackled? Furthermore, there might be one or more lurking variables for which a dependency structure can be observed. Some discussion in this direction is necessary.

A4: We have calculated the variance inflation factor and the Pearson correlation between the explanatory variables in the models presented in Table 3. A typical consequence of multicollinearity is the instability of parameter estimates and the solution is to remove those covariates that create a dependence structure. Here, because VIF is all less than 5 and the correlation between variables is not strong, the influence of multicollinearity is small. We preferred to present the models with all covariates in order to compare the different results when addressing different types of near-miss events.

Q5: Finally, are all the assumptions and/or necessary and sufficient conditions to conduct a Poisson and/or Negative binomial regression are met?

A5: We have included the following sentence in the Methods section from Line 200 to Line 204.

“Our observed drivers can be considered independent of each other. Even if they drive in a similar area, they do not have any apparent relationship with each other. When we observe one driver over time, we have taken care of temporal correlation using the panel model that considers that one individual is observed repeatedly, here each day.”

Q6: Page 7, Table 3: For each regression (best-fitted model), except for Negative binomial regression, nothing is mentioned on the efficacy of the slope coefficients! The overall goodness of fit is not sufficient enough, we need to examine the role of each predictor. Some discussion in this direction is necessary.

A6: We agree that Table 4 (old Table 3) only shows the goodness of fit. It was aimed at showing that the Negative binomial model provides the best fit for each dependent variable. So, we show and discuss the results in Table 5, A1, A2, and A3 in the first paragraph of the Discussion.

We truly appreciate the questions suggested. They have given us new ideas for further research that we will certainly explore in the future.

Best Regards,

Shuai Sun, Jun Bi, Montserrat Guillen, Ana M. Pérez-Marín

Reviewer 2 Report

I have the following comments:

  • Give some numerical results in Abstract.
  • In keywords explain what is GLM.
  • There are no research questions and no hypotheses. Some of them should be added to the text because the research aim is not sufficient.
  • I suggest dividing the paper into the following parts: Introduction, Literature review, Methods, Results and Discussion, Conclusions.
  • Please add the difference between this work and the previous studies in literature.
  • Give more results in Conclusion.
  • Give more explanations for Fig.2 and explain the steps of the methodology.
  • Data for one month were taken into account. How the influence of the season is taken into account - winter, autumn. The data are for the summer season. What are the seasonal fluctuations?
  • The discussion needs to be expanded. Explain the practical usefulness of the models and how they will be used. An example of a risk assessment should be added, and also an example for use the results in insurance.

Author Response

Dear reviewer,

Thank you for your constructive suggestion for our flawed manuscript. We have undertaken major revisions and taken all your precious comments into consideration. Please find our detailed replies to the specific comments below.

Q1: Give some numerical results in Abstract.

A1: Regression analysis was carried out on the summary data and panel data respectively and four risk events were used as dependent variables. Too many numerical results would make the Abstract not concise enough, but we still added a few numerical descriptions. Here is the new Abstract.

“This study proposes a method for identifying and evaluating driving risk as a first step towards calculating premiums in the newly-emerging context of usage-based insurance. Telematics data gathered by the Internet of Vehicles (IoV) contain a large number of near-miss events which can be regarded as an alternative for modeling claims or accidents for estimating a driving risk score for a particular vehicle and its driver. Poisson regression and Negative binomial regression are applied to a summary data set of 182 vehicles with one record per vehicle and to a panel data set of daily vehicle data containing four near-miss events i.e., counts of excess speed, high speed brake, harsh acceleration or deceleration and additional driving behavior parameters that do not result in accidents. Negative binomial regression (AIC_overspeed=997.0, BIC_overspeed=1022.7) is seen to perform better than Poisson regression (AIC_overspeed=7051.8, BIC_overspeed=7074.3). Vehicles are separately classified to five driving risk levels with a driving risk score computed from individual effects of the corresponding panel model. This study provides a research basis for actuarial insurance premium calculations, even if no accident information is available, and enables a precise supervision of dangerous driving behaviors based on driving risk scores.”

Q2: In keywords explain what is GLM.

A2: GLM stands for Generalized Linear Model, which has been changed to its full name in keywords.

Q3: There are no research questions and no hypotheses. Some of them should be added to the text because the research aim is not sufficient.

A3: The first three paragraphs of Introduction all describe the research aim of this paper, and we also add more words in the fourth paragraph of Introduction and the last paragraph of Methods to supplement the explanation.

Q4: I suggest dividing the paper into the following parts: Introduction, Literature review, Methods, Results and Discussion, Conclusions.

A4: We have adopted the suggestion to separate the Conclusion from the Discussion, but we still believe that the Data description should be a separate section, because this study is based on data, so it is necessary to clearly provide detailed description of the data.

Q5: Please add the difference between this work and the previous studies in literature.

A5: The following text was added to the final paragraph of the literature review.

“Compared with previous studies, this study not only conducts regression on the summary data set to model and analyze the factors causing risk events, but also conducts panel data regression on the panel data set to consider individual effects and time effects. The regression results can not only make more accurate causal inference, but also carry out risk scoring.”

Q6: Give more results in Conclusion.

A6: More content has been added to the Discussion as well as to the Conclusion.

Q7: Give more explanations for Fig.2 and explain the steps of the methodology.

A7: The corresponding content is added to the last paragraph of Methods.

Q8: Data for one month were taken into account. How the influence of the season is taken into account - winter, autumn. The data are for the summer season. What are the seasonal fluctuations?

A8: It is a good suggestion that we can consider the influence of seasonal fluctuations of long-term data on the modeling results in future studies. However, the purpose of this study is to evaluate the driving risk of the observed values in the condition of short data collection period. In traffic practice, it is easier and more reliable to track short-term data than long-term data to avoid external changes that may affect the driver’s routines, such as place of residence or job for example.

Q9: The discussion needs to be expanded. Explain the practical usefulness of the models and how they will be used. An example of a risk assessment should be added, and also an example for use the results in insurance.

A9: The role and usage of the risk assessment model of this study are stated in the first paragraph of Discussion and the third paragraph of Conclusion. The whole Results section and the Discussion section are an example of a complete driving risk assessment. The assessment results are shown in Figure 4. The driving risk level can be converted into a risk factor and introduced into the calculation process of car insurance rate determination. This work will be further studied and discussed in future studies.

We truly appreciate the questions suggested. They have given us new ideas for further research that we will certainly explore in the future.

Best Regards,

Shuai Sun, Jun Bi, Montserrat Guillen, Ana M. Pérez-Marín

Reviewer 3 Report

Dear authors,

Please see attached my review.

Author Response

Dear reviewer,

Thank you for your constructive suggestion for our flawed manuscript. We have undertaken major revisions and taken all your precious comments into consideration. Please find our detailed replies to the specific comments below.

Q1: Line 26: IoV must be defined as it is the first appearance in the text, after the Abstract.

A1: The full name of IoV for the Internet of Vehicles, has been added to its first appearance.

Q2: Line 66: The sentence “The rest of this article...” must start a new paragraph.

A2: Introduction has been segmented as you suggested.

Q3: All formulas used in equations (1)- (5) must be defined.

A3: The corresponding definition content has been added according to your suggestion.

Q4: All figures must have sources.

A4: All figures are generated from the data and results of this study or are self-drawn without attribution of source. If someone is quoting it, however, it should be attributed to this article.

Q5: There is a big gap between Tables 4 and 5.

A5: This is a problem with LaTeX typesetting, which has been eliminated through adjustment. We will work with the editor to improve the display.

Q6: The authors must explain how they select the 182 vehicles for which they collected the data and then analyzed.

A6: The research data were collected and provided by China Satellite Navigation and Communications Co., Ltd. The data of 182 vehicles were the entire data of one data set. The methods and details of data processing are shown in the indexed literature. We also added the following in the first paragraph from Line 141 to Line 143 of the Data Description.

“Although we cannot obtain more data due to the commercial privacy of the data, the limited data also contains valuable driving risk information, which is worth studying.”

Q7: The authors should mention the novelty of the paper in the Introduction section.

A7: The novelty of the paper is introduced in the second paragraph of the Introduction.

Q8: A table with all variables used in the model and their definitions and unit measure would be useful.

A8: Relevant contents are included in Table 1 and the Abbreviations.

Q9: Limitations of the study must be added in the Conclusions section.

A9: The limitations of the study are expressed in detail in the first and second paragraphs of the Conclusion.

Best Regards,

Shuai Sun, Jun Bi, Montserrat Guillen, Ana M. Pérez-Marín

Round 2

Reviewer 1 Report

The authors must be thanked for addressing all my major concerns with thorough details. I recommend this paper for a possible publication after minor editing for the English language and grammatical errors/typos, etc. 

Reviewer 2 Report

The authors have taken into account all the remarks given in the review. Wherever there are citations, there are question marks. Maybe this is some formatting error. The article has been improved and can be published after replacing the question marks with the real sources.